# First Steps Towards Site Characterization Activities at the CSTH Broad-Band Station of the Campi Flegrei’s Seismic Monitoring Network (Italy)

**DOI:** 10.3390/s25154787

**Published:** 2025-08-03

**Authors:** Lucia Nardone, Rebecca Sveva Morelli, Guido Gaudiosi, Francesco Liguoro, Danilo Galluzzo, Massimo Orazi

**Affiliations:** 1Istituto Nazionale di Geofisica e Vulcanologia (INGV), Osservatorio Vesuviano, Via Diocleziano 328, 80124 Napoli, Italy; lucia.nardone@ingv.it (L.N.); guido.gaudiosi@ingv.it (G.G.); francesco.liguoro@ingv.it (F.L.); danilo.galluzzo@ingv.it (D.G.); massimo.orazi@ingv.it (M.O.); 2Dipartimento di Scienze della Terra, dell’Ambiente e delle Risorse (DISTAR), Università degli Studi di Napoli Federico II, Via Vicinale Cupa Cintia 21, 80126 Napoli, Italy

**Keywords:** seismic site characterization, CSTH seismic BB station, spectral analysis

## Abstract

Local site conditions can significantly influence the amplitude, duration, and frequency content of seismic recordings, making the characterization of subsoil properties a critical component in seismic hazard assessment. However, despite extensive research, standardized methodologies for assessing site effects are still lacking. This study presents preliminary steps in the site characterization of a small area of Campi Flegrei caldera (Italy), with the aim of enhancing understanding of local lithology and seismic wave propagation. The analysis focuses on the broad-band seismic station CSTH, installed in 2021, and incorporates data from a temporary 2D array of five short-period sensors deployed around the station. These sensors recorded both ambient noise and seismic events associated with caldera dynamics. To improve the robustness of the characterization, data from two additional permanent broad-band stations (CPIS and CSOB) of the Istituto Nazionale di Geofisica e Vulcanologia—Osservatorio Vesuviano’s monitoring network, also located nearby a hydrothermal field, were included. Spectral analyses such as Power Spectral Density (PSD), Horizontal-to-Vertical (H/V) spectral ratios, and f-k array technique were performed to evaluate the frequency-dependent response of the site and to support the development of a comprehensive seismic site model.

## 1. Introduction

Seismic site effects play a crucial role in modulating the amplitude, duration, and frequency content of ground motion and are therefore critical to seismic hazard assessments. Numerous studies have investigated local site effects and their characterization [1,2,3,4]. Traditionally, site characterization is achieved through the estimation of local S-wave velocity profiles and by identifying the fundamental resonance frequency of near-surface layers [5]. A commonly used parameter in microzonation studies is the average shear-wave velocity in the top 30 m (Vs30); however, several studies have highlighted the limitations of Vs30-based classification, particularly in complex geological and topographical contexts, where slope-derived proxies may misrepresent site conditions, such as in volcanic terrains or sedimentary basins with low relief, and may suffer from regional biases due to calibration constraints and the inability to capture subsurface heterogeneities or basin structure [6].

In response to these limitations, passive seismic methods have gained increasing popularity due to their non-invasive nature and cost-effectiveness. Among these, techniques based on ambient noise recordings—such as single-station Horizontal-to-Vertical (H/V) spectral ratio methods [7] and 2D array approaches for surface wave dispersion analysis [1,8,9] have demonstrated significant potential. Notably, Parolai et al. [4] showed a strong agreement between H/V ratios derived from ambient noise and earthquake recordings, particularly for estimating the fundamental frequency of sedimentary layers.

Advancements in joint inversion techniques, such as those developed by [5] using H/V and surface wave dispersion curves (e.g., ESAC by [10] and f–k analysis by [11]), have enabled more robust S-wave velocity profiles, even in complex urban settings. Furthermore, recent studies have explored the temporal variability of H/V ratios as indicators of subsurface changes, including permafrost dynamics [12], rock mass stability [13], and seasonal variations in mechanical properties [14]. Similarly, La Rocca et al. [15] and Vassallo et al. [16] demonstrated that H/V temporal changes can be associated with shallow velocity variations, especially following strong seismic events.

Additional contributions by Felicetta et al. [17] and Cultrera et al. [18] have emphasized the importance of standardizing site characterization procedures across national seismic networks to improve data quality and utility.

Within this framework, we present the preliminary site characterization of the digital broad-band seismic station CSTH, operated by the Istituto Nazionale di Geofisica e Vulcanologia—Osservatorio Vesuviano (INGV-OV). CSTH is part of the permanent seismic monitoring network of the Campi Flegrei caldera in southern Italy. This is one of the most hazardous volcanic areas globally due to its active magmatic system, dense population, and history of large-magnitude explosive eruptions [19].

The present study is part of the PON project “Geoscience Research INfrastructure of ITaly” (GRINT), which aims to enhance access to high-quality geophysical data and improve monitoring strategies for seismogenic and volcanic processes in southern Italy (https://progetti.ingv.it/en/pongrint, accessed on 20 June 2025). As part of this initiative, INGV-OV is conducting subsurface characterizations of the reference seismic station to refine our understanding of the local soil response and improve ground-motion prediction models [20].

Here, we describe the deployment of a temporary 2D array of five short-period seismic stations around CSTH that recorded both ambient noise and local seismicity for approximately three days. Additional data from two nearby permanent stations (CPIS and CSOB) were integrated into the analysis. Despite challenges posed by anthropogenic noise in this densely populated area, the collected dataset enabled spectral analyses, including Power Spectral Density (PSD) estimation, Nakamura’s H/V spectral ratio method [21,22,23], and f-k array technique [24], to assess site resonance, wavefield characteristics, and derive the Rayleigh wave dispersion curve.

## 2. Geological Characterization

CSTH station is located in the central part of the Campi Flegrei caldera (Figure 1), considered one of the world’s highest volcanic risk areas [25]. The caldera is marked by a ground deformation characterized by alternating phases of uplift and subsidence known as bradyseisms [26]. This deformation is accompanied by intense seismic activity [27] and significant fluid emissions [28], especially in the Solfatara and Pisciarelli areas (Figure 1b). The caldera has been experiencing a new phase of volcanic unrest since 2005, marked by ground uplift and increasing seismicity [29], with recent deformation rates exceeding 18 cm/year.

The geology of the entire area is dominated by two calderic structures that help form the present-day depression of Campi Flegrei, within which there are numerous tuff cones, ring tuffs, and minor lava domes. The area has also been affected by multiple marine ingressions in the past. Rock outcrops are composed of continental and marine sediments interbedded with pyroclastic deposits, mainly derived from pyroclastic density currents in proximal areas and pyroclastic fallout in distal areas [30]. These deposits are composed of loose pyroclastic material (ash, pumice, slag, lithics) or lithified tuffs and minor lavas and, in lowland areas, are intercalated with marine and coastal sediments. Figure 2 shows the detailed geological map of CSTH’s surrounding area, extracted by Isaia et al. [31], in which the superposition of volcanic deposits put in place by the numerous volcanic eruptions that have occurred in the area is clearly evident [32]. The reconstructed geological section along the A-A’ profile (top right panel in Figure 2) shows a layered geometry of the subsoil down to a 100 m depth and highlights the presence of a complex fault system that dislocates deposits attributable to the Astroni, Averno, Agnano-M. Spina, and Paleoastroni eruptions (Figure 2). These deposits, put in place during the most recent activity of the central sector of the caldera, are horizontally layered and mainly composed of coarse and fine pyroclastic materials, often deeply altered by secondary mineralization due to fumarolic and hydrothermal activity [31].

## 3. Specifications of the Broad-Band Station “CSTH” and the Temporary Array

The CSTH digital seismic station (http://orfeus-eu.org/stationbook/networks/IV/1988/stations/CSTH/2021/; last access, 2 August 2025) is currently housed inside a suitably insulated pit in a private garden, 200 m from the Pisciarelli fumarole (Figure 1b). The station, installed in June 2021 alongside the analog STH station, is equipped with an Affinity digitizer (Guralp, Figure 3a), a 120 s Guralp 3ESPC very broad-band velocimeter sensor (Figure 3b), and a Guralp Fortis accelerometer sensor (from Güralp Systems Ltd., Reading, United Kingdom; https://www.guralp.com/products, last access 2 August 2025; Figure 3c).

The other stations of the Campi Flegrei monitoring network installed in the area, and used in this study, are CPIS station (http://orfeus-eu.org/stationbook/networks/IV/1988/stations/CPIS/2010/, last access 2 August 2025), located a few meters from the emission point of the Pisciarelli fumarole, and CSOB station (https://orfeus-eu.org/stationbook/networks/IV/1988/stations/CSOB/2021/, last access 2 August 2025/), located on the western crater rim of the Solfatara volcano. CSOB station has the same instrumental configuration as CSTH station, while CPIS station is equipped with a Gilda digitizer [33] and a Guralp CMG-40T-60S. The temporary stations, located within a 300 m radius from the CSTH station, identified as STH*, were installed according to a geometry that necessarily had to take into account the logistics of the area (Figure 1b). Specifically, temporary station STH1 was installed within the garden of the Altamira Complex; temporary station STH2 was installed next to the soccer fields immediately downstream of the Pisciarelli fumarolic area; temporary station STH3 was installed within private property; temporary station STH4 was installed on private land; and temporary station STH5 was installed within a flowerbed at Ariel Car Pozzuoli. All STHs are equipped with a Lennartz 3DLite (https://reftek.com/lennartz-sensors; last access 2 August 2025) and either Lunitek Atals (STH1, STH2, and STH4; https://lunitek.it/seismic/seismic-recorders/atlas/; last access 2 August 2025) or Guralp Affinity (STH3 and STH5) digitizers. The CSTH, CPIS, and CSOB seismic stations continuously transmit data to the INGV-OV acquisition center, whereas the temporary stations locally record data in memory cards. Synchronization at each station was achieved using the GPS time signal and data were digitized at 100 sps. The information about the temporary stations can be found within the attached data sheets in the appendix. The difficulties faced with the installation of the five temporary stations mainly concerned the choice and accessibility of the sites identified during the design phase. Given the presence of man-made infrastructure, it was necessary to find the right compromise between the desired geometry and the accessibility and availability of the identified sites.

The instrumental characteristics and coordinates of the temporary array are shown in Table 1. Recordings from this array (orange circles in Figure 1b) were integrated with data from the broad-band stations of the permanent monitoring network of Campi Flegrei (blue circles in Figure 1b) for a better characterization of the area. Figure 4 shows the acquisition period (chronogram), where the green bands show the period of recording while the red bands indicate the absence of acquisition. No gaps during the operating time are present in the dataset.

The choice and selection of the sites for the installation of temporary seismic stations was made taking into consideration the following key aspects: the willingness of private citizens to host the instruments and safe accessibility to the site.

Usually, the process for the array installation consists of a preliminary laboratory selection of individual sensor sites, dependent on the characteristics and resolution of the entire array (see Figure 5). In this case, however, given the heavy urbanization and extensive human modification of the entire Campi Flegrei area, it was first necessary to verify in situ which areas were most suitable and accessible for sensor installation in the vicinity of the CSTH station. Only once the possible installation areas have been identified did we proceed to verifying the resolving limits according to the different possible geometries. Finally, once the best geometry was chosen, it was necessary to obtain approvals from the owners of the individual areas.

### Array Resolution

Figure 5a shows the geometry of the installed array together with the three permanent monitoring stations, and Figure 5b shows the resolution curves corresponding to kmin (thick dashed black line), kmax (solid black line), and kmax/2 = kmin/2 (dotted black line). “Warangps” (https://www.geopsy.org/man/warangps.html, last access 20 June 2025) from the Geopsy package [34] was used to determine the array geometry limits. Indeed, the minimum and maximum resolvable wavelengths, kmin and kmax, respectively, are directly related to the array geometry (minimum inter-station spacing and maximum aperture) and were estimated from the array transfer function. The survey depth depends on the maximum measured wavelength, and the resolution decreases with depth. In particular, the velocity values of the dispersion curve obtained at high frequencies determine the resolution of the model at its most superficial part (small wavelengths) [9]. Taking into account the array resolution limits and the vs. velocities obtained by [9] in the central part of the caldera, we expect that we will be able to obtain a dispersion curve in the 1.0–4.0 Hz frequency band.

## 4. Data Analysis

To achieve high-quality and reliable site characterization of the CSTH station we followed the steps outlined below:-First, we made a spectral analysis (Spectrograms and PSDs) of the entire dataset, including both seismic noise and earthquakes, to highlight possible variation from site to site. In general, the properties of seismic noise depend on proximity to populated areas or the coastline, local earthquake rate, local average wind speeds, and geological features.-In the second stage, we have computed spectral ratio (H/V) to assess whether the area around CSTH exhibits a homogeneous response without any significant local site effects or not. This is a basic starting point to obtain a reliable and representative velocity model of the site when seismic array techniques are applied.-The f-k technique was applied to the vertical component of ground motion only after confirming that the conditions of the first two processing stages were met, in order to derive the Rayleigh wave dispersion curve.

All analyses were performed through the use of two packages: Geopsy [34] and ObsPy [35]. Geopsy, created in 2005 during the European SESAME project, has provided tools for processing environmental vibrations with the goal of characterizing sites; progressively more conventional techniques (such as MASW or refraction) have been included to provide a high-quality, comprehensive, and free platform for interpreting geophysical experiments (https://www.geopsy.org/, last access 20 June 2025). Obspy (https://docs.obspy.org/, last access 20 June 2025), a tool in Phyton https://doi.org/10.5281/zenodo.15298023., is an open-source library for seismological analysis and is designed to access and process many formats in seismology.

### 4.1. Dataset

Our dataset [36] consists of almost 3 days of seismic noise and earthquakes collected from eight stations. Figure 6a shows, by way of example, the seismic traces in the three vertical (V) and horizontal (E, N) components of nearly 10 h of recording on 28 May 2022 (02:40:00 to 12:00:00 UTC) of the five temporary stations (STH*) and the three stations of the permanent Campi Flegrei network (CSTH, CSOB, and CPIS). The highest signal amplitude is observed on the vertical component of station STH5. The higher amplitude observed at this station could likely be explained by a combination of local site effects and anthropogenic noise. The station is installed on softer, unconsolidated ground typical of urban areas, and its proximity to a busy road introduces continuous traffic-induced vibrations. These factors can amplify seismic signals and influence their amplitude and shape. Overall, the recorded amplitude increase is attributed to these environmental and geological conditions, rather than any instrumental or data processing issue.

During the array acquisition period, 18 seismic events (*M_d_* between −0.5 and 0.7) were recorded and localized in the Campi Flegrei area [37]. Figure 6b shows the seismic traces of two seismic events: the first has *M_d_* 0.2, which occurred at 22:23:02 UTC, and the second *M_d_* 0.7, which occurred at 22:23:12 UTC on 28 May 2022. The more energetic event (*M_d_* 0.7) was located at a depth of 0.7 km a.s.l. within the crater of Solfatara Volcano (Figure 7). The station CSTH is regularly used to locate seismic events and, in addition being the reference station for Campi Flegrei, is installed between Solfatara and Pisciarelli in the area where the largest part of the seismicity (shallow and low in magnitude) is located [27]. This makes CSTH one of the most important stations in the area and highlights the need for detailed site characterization to improve data quality and to identify and interpret potential signal anomalies.

The amplitude recorded at STH1 station, in terms of both seismic noises and earthquakes, is lower than that for the other stations. This is most likely due to an instrumental failure rather than a site-related effect.

With the aim of analyzing the frequency content of the two selected earthquakes, we calculated the frequency spectra of the seismograms in the 1–40 Hz range after converting the traces from count to velocity (m/s). Spectral amplitude values were color-coded, with red representing low values, ranging from yellow and green to blue for the high values. Figure 8 shows the spectrograms of the two seismic events relative to the vertical components. The spectrograms are typical of VT (volcano-tectonic) earthquakes with an impulsive onset characterized by a wide frequency range up to 40 Hz. Indeed, the maximum frequency content for all stations is between 10 and 40 Hz, with a low-frequency component, between 4 and 10 Hz, lasting up to 10 s in the case of the second event. This frequency pattern is characteristic of shallow low-energy VT earthquakes.

### 4.2. Estimation of the Power Spectral Density (PSD)

The main indicator of the “performance” of a seismic station and also an indicator of the “quality” of a site is the level of recorded seismic noise. Anthropic activity, but also ocean microseisms, temperature and pressure fluctuations, and instrumental noise from the sensors and dataloggers are among the main sources of seismic noise. Placing the station away from human infrastructure, selecting sites with suitable lithology, ensuring thermal and pressure insulation, and optimizing ground coupling can significantly reduce the level of noise recorded by a seismometer. Background noise can also vary over time due to changes in the nearby human activity, and continuous noise monitoring enables network managers to assess station performance over time.

Power Spectral Density (PSD) provides a quantitative analysis of the spectral power distribution over an entire frequency range by integrating the probability density function in the definition of a specific random process [38]. It is a standard method for quantifying seismic background noise. The main objective of PSD analysis is to estimate the spectral density of data recorded by seismic stations, which is achieved by computing the Fourier transform (FT) of the autocorrelation function of the signals. PSDs are the most commonly used tool [39] to obtain a picture of how the noise level is distributed in amplitude and frequency during the entire time interval inspected (weekly, monthly, or yearly).

In order to characterize the background noise at and around CSTH station in terms of spectral amplitude, check its stability over time, and compare it with standard reference noise models [40], PSDs were computed for the three components of motion at all stations over the entire acquisition period (Figure 9). The PSDs and their probability density functions (PDFs) were obtained from the noise waveform after the convolution of the instrumental response functions using ObsPy (1.4.2 version) package [35]. For each seismic channel, the software calculates the PDF of the distribution of the PSD values at each frequency bin. The PDFs represent the occurrence probability of a given seismic signal level within a specific frequency interval.

In the different panels of Figure 9, each colored curve refers to the spectrum calculated using a different timepoint of the analyzed period, while the grey curves represent the reference New Low Noise Model (NLNM) and New High Noise Model (NHNM) [40]. In general, it can be observed that for all stations, for periods shorter than 1 s, particularly for periods below 0,4 s (where the effect of road traffic is most pronounced), PSD levels approach and/or exceed the NHNM reference curve. As was to be expected in densely urbanized areas, despite efforts to mitigate anthropogenic noise, all sites are found to be “very noisy”. Notable differences in noise levels between day and night, as well as during the weekend, were observed. The spectral amplitudes of the five temporary stations show asymptotic behavior for periods greater than 1 s. This is due to the type of sensor used for the measurements (Lennartz 3DLite 1s), which limits the ability to characterize the spectral amplitudes in that period (or frequency) band.

The decrease in spectral amplitudes is again related to instrument characteristics, especially the sensitivity of the broad-band sensors.

A peak between 1 and 2 s is present in all spectra (except that from station STH1) that is evident on both the vertical and horizontal components (Figure 9). This peak, characteristic of marine microseisms, in the Campi Flegrei Caldera appears to be systematically shifted [41] from the maximum reference value (NHNM), which typically occurs between 3 and 6 s. In the PSDs of CSOB broad-band station at long-period ranges of more than 20 s, the horizontal components of noise lie significantly above the NHNM compared to the vertical component. This is probably due to a local effect [42] or surface displacements caused by thermal and barometric variations. In this period range, which is of great importance in seismic source studies, background noise is sensitive to diurnal and seasonal temperature variations.

Among the broad-band stations, the PDFs of the CPIS vertical component lie significantly above the NHNM in the 20–70 s period range and below 0.1 s. This station is located approximately 10 m from Pisciarelli fumarole, where a persistent harmonic tremor is present [43].

### 4.3. Spectral Ratio Analysis

The Horizontal-to-Vertical Spectral Ratio technique [21,22] was applied to the noise recordings with the aim of estimating the fundamental resonance frequency of the installation sites. Vassallo et al. [16] provides a detailed explanation of how this technique can be used to retrieve information about the seismic properties of shallow subsoil via single-station measurements. The “Nakamura method” aims to estimate the site resonance frequency and is typically applied to ambient noise recordings (microtremors) [28]. Ambient seismic noise is recorded at the surface using a three-component seismometer. The horizontal components are geometrically averaged to obtain a representative horizontal motion, while the vertical component is used directly. The spectral ratio is then computed as the amplitude spectrum of the horizontal component divided by that of the vertical one. The resulting HVSR curve reveals characteristic frequency peaks, which are interpreted as the site’s fundamental resonance frequency (F_0_), while the amplitude of the peak is considered indicative of the site amplification factor. To ensure signal quality, long-duration recordings are recommended (typically exceeding one hour) and transient disturbances, especially those related to anthropogenic activity, should be removed. This is achieved by segmenting the signal into moving time windows (commonly 250 s in length) with a standard overlap of 10% and applying a “detrigger” or “antitrigger” algorithm to exclude non-stationary segments. Each selected window is processed using a Fast Fourier Transform (FFT), and the resulting spectra are smoothed using the Konno–Ohmachi smoothing algorithm, with a bandwidth coefficient (b) commonly set to 40. A band-pass filter (e.g., 0.4–20 Hz) is typically applied to remove low-frequency drift and high-frequency noise prior to spectral analysis. The final H/V curve is derived as the average of the spectral ratios computed for all accepted windows. The dominant peak in this curve is identified as F_0_, which corresponds to the resonance frequency associated with the shallow subsurface layers. The peak amplitude is taken as a proxy for the amplification factor due to impedance contrast across the stratigraphy [28,43]. Our analysis was conducted on approximately 10 h of recordings on 28 May 2022, from 02:40:00 h to 12:00:00 UTC. After several tests, in order to remove the effect of transient disturbances in the signals attributable to non-stationary anthropogenic sources located in the vicinity of the stations, a “detrigger” algorithm [44] was applied to a total of 1296 moving signal windows; then, the procedure described above was applied. The Fourier spectra of the NS and EW components were geometrically averaged to obtain the Fourier spectrum of the horizontal component. For the windows selected by the “antitrigger” algorithm (parameters are given in Table 2), smoothing was applied to the calculated Fourier spectra according to the Konno–Ohmachi algorithm [45], setting a value of *b* equal to 40. The final H/V curves were then calculated using the geometric mean of the spectra of the horizontal components as the numerator and the spectrum of the vertical component as the denominator.

The automatic H/V on the data was performed, potentially including both noise and earthquakes; although the H/V on noise is often similar to that for earthquakes, sometimes significant differences have been observed [46,47]. Nevertheless, since no notable earthquakes were recorded on the days covered by the analysis (the highest *M_d_* recorded was 0.7), this difference in this case may be negligible. In any case, Vassallo et al. [16], for example, compared the automatic H/V results with those obtained through a manual selection of data and found that the proposed automatic H/V analysis provides very similar results to the manual analysis, regardless of earthquakes and transients.

Figure 10 shows the average H/V curves (black solid line) with their standard deviations (dashed black lines) and the H/V curves obtained on each sliding time-window (colored thin lines). Figure 11, on the other hand, shows the superposition of the average H/V curves obtained at all stations on the left (Figure 11a) and their average on the right (Figure 11b).

The H/V curves generally exhibit a flat and consistent trend (Figure 10 and Figure 11a), with spectral amplitudes rarely exceeding a value of 2. This behavior suggests the absence of significant impedance contrasts in the subsurface within the analyzed frequency band that could produce relevant amplification effects. The only exception is the CSOB station, which displays a peak centered around 15 Hz, with an amplitude slightly above 2. This anomaly can likely be attributed to the presence of contact between loose pyroclastic deposits and more compact layers at the station site.However, the low amplitude and relatively high frequency of the peak indicate a weak near-surface impedance contrast, probably associated with a limited and localized stratigraphic heterogeneity.

#### Temporal Representation H/V Profiles

The entire dataset has been divided into 16 consecutive four-hour-long intervals (Figure 12 right site, legend with analysis window intervals), and the H/V over each time interval has been calculated, using the same parameters described in Section 4.3. The temporal variations of averaged H/V curves, computed among the single four-hour interval as a function of frequency and time, are plotted in Figure 12 in order to verify their stability over time. The maximum amplitudes were normalized to the color scale, allowing us to see possible amplitude variations from site to site. The white bands are due to the absence of seismic data; station STH2 has fewer data points to represent due to technical instrumentation problems.

The highest amplitude in the spectral ratios is evident at station CSOB (~1.30–2.70) in the 1–6 Hz frequency band, probably due to strong localized disturbance in the vicinity of the station. Indeed, this highest amplitude begins on the second day of the experiment and remains until the end. None of the other stations exhibit this distinctive feature. Smaller amplitudes are recorded instead at the CPIS station (~0.35–1.10) due to the contribution of fumaroles and the mud/water pool; in fact, at the CPIS station there is a strong tremor due to gas leakage and its polarization is almost in the vertical plane [43]. Therefore, the amplitude of the H/V ratio is minimal. Conversely, the other stations exhibit approximately similar ratios (~0.90–1.55), with the ratio being 1–2. At STH4 station we see an anomaly for a single window (29 May 2022 from 08:00 to 12:00 UTC) that we do not record at the other stations, probably due to a local disturbance in the vicinity of the station.

### 4.4. Rayleigh Wave Dispersion Curve and Velocity Model

Assuming that the Rayleigh waves dominate the wavefield generated by vertical point sources acting on the surface around the array, we applied the f-k technique [24] to our data in order to extract the characteristics of the Rayleigh wave propagation in the subsoil (i.e., the Rayleigh wave dispersion curve). Indeed, surface waves implicitly carry the information related to the medium they are travelling through and hence, if correctly analyzed, their propagation can be a useful site characterization tool.

The f-k analysis was performed using 11 h of the vertical component of the seismic noise within the 1–7 Hz frequency band, with a 0.1 Hz central frequency (fc) step and using 100 s sliding time windows. The minimum frequency was chosen on the basis of the array resolution limits. The values associated with all the selected time windows at the fixed frequency were used to calculate the mean slowness and its associated statistical error [48], resulting in the creation of a histogram (Figure 13). The phase velocity dispersion of the fundamental mode of Rayleigh waves was obtained by plotting the inverse of slowness as a function of frequency. The manually picked dispersion curve, corresponding to the black continuous line with vertical error bars in Figure 13, was plotted within the area bounded by the array resolution limits (Figure 13). The dispersion pattern defined in a limited frequency band is strongly correlated to the array configuration. Figure 13 shows the phase velocity ranging from about 1100 m s^−1^ at a frequency of 1.3 Hz to about 600 m s^−1^ for frequencies between 4 and 5 Hz.

We performed an initial inversion of the dispersion curve to obtain a preliminary shear-wave velocity model (Figure 14), which, although still in an early stage, provides meaningful constraints for the seismic characterization of the site. This model offers insights into the vertical velocity distribution of the shallow subsurface layers, contributing to a better understanding of local stratigraphy and mechanical properties relevant to site response analysis. The resulting velocity model, composed of two seismic layers overlaying a half-space, is consistent with the models obtained in the Campi Flegrei area by [9,49,50]. Here, the inverted velocity models were interpreted as indicative of loose pyroclastic deposits and altered facies of the Neapolitan tuffs in the shallow subsurface. Similarly, the results obtained at the CSTH station, when compared with the geological cross-section A-A’ in Figure 2, suggest that the subsurface is composed of the same lithological units, supporting the hypothesis of a laterally continuous stratigraphic setting around the station, at least at the scale of investigation. To assess the robustness of the solution, nine inversion runs were performed, each exploring 15,050 models, for a total of 135,000 models analyzed. The results were projected onto the Vs–h parameter space (Figure 14d), as these parameters exert the strongest influence on the dispersion curve. In the figure, each point corresponds to a tested model, colored according to how well it fits the observed data. Overall, all parameters converge well toward the final solution, demonstrating that the inversion process was both reliable and stable.

## 5. Conclusions

This work provides a preliminary yet essential contribution to the seismic site characterization of the CSTH broad-band seismic station, strategically located in the central sector of the Campi Flegrei caldera, one of the most hazardous and densely populated volcanic areas in the world. By integrating passive seismic techniques, including ambient noise analysis, H/V spectral ratio, and f-k analysis, this study offers a robust evaluation of local site effects and a multi-scale characterization of ambient seismic noise, despite the challenges posed by high anthropogenic noise levels [51]. In particular, this integrated approach supports both local-scale analysis, using single-station methods such as H/V and PSD, and broader spatial investigations via array-based techniques like f-k analysis, which exploit the geometric distribution of stations to extract additional information on the direction and phase velocity of surface waves. This methodological synergy is especially effective in a geologically complex setting like the Campi Flegrei area, where the heterogeneity of volcanic deposits and ongoing seismic–volcanic activity demand tools capable of accurately capturing both the local site response and the seismic properties of the shallow subsurface. The results reveal a relatively homogeneous seismic response across the investigated area, with no significant resonance peaks in the H/V ratios. Although our study does not reveal significant amplification effects related to subsurface stratigraphy, very high peak ground acceleration (PGA) values have been recorded in the Campi Flegrei area. The observed values are extremely variable and highly localized near the epicentral region; this variability can be partially attributed to the location and depth of the seismic event (https://www.ingegneriasismicaitaliana.com/articoli-tecnici/ingv-isi-campi-flegrei-nuove-prospettive-nella-valutazione-del-rischio-sismico-attraverso-approcci-integrati-di-monitoraggio-sismico-e-strutturale, Italian site, last access 20 July 2025). The inverted velocity model is consistent with layered pyroclastic deposits and altered volcanic materials consistent with previous geological and geophysical studies [9,31] focused on this volcanic area. Here, we demonstrate that, despite using sensors with different characteristics, a consistent objective can be achieved, provided that the calibration is rigorous and up-to-date, the instrumental responses are known and correct, and internal (manual and automatic) data quality checks are performed. Hauksson et al. [52], for instance, demonstrate that reliable seismic site characterization can be achieved using sensors with different technical characteristics, such as broad-band seismometers, short-period sensors, and strong-motion accelerometers, provided that proper calibration and metadata validation are applied. Despite the diversity in sensor types, consistent ground motion recordings and parametric results were obtained, confirming the feasibility of integrating heterogeneous instruments for high-quality seismic analysis. Earlier, Endrun et al. (2010) [53] also demonstrated that their measurements, made using different sensors and under varying conditions, produce consistent dispersion and autocorrelation curves, yielding stable information over time.

Our study was conducted during the ongoing bradyseismic crisis, marked by accelerating ground uplift and increased shallow seismicity. In this context, accurate site characterization is a strategic priority, as it enhances assessments of local amplification effects and refines subsurface velocity models critical for real-time monitoring. CSTH station, located between the main degassing zones of Solfatara and Pisciarelli [54], plays a vital role as a reference point for both earthquake localization and magnitude estimation.

Effective risk mitigation hinges on a thorough understanding of local site conditions, which is key to improving seismic hazard assessments and informing civil protection strategies. In a densely urbanized caldera like Campi Flegrei, where both human exposure and infrastructure vulnerability are high, understanding how local geology influences seismic wave propagation is essential.

Future work will involve expanded data acquisition and the application of joint inversion techniques to improve the resolution of the shallower part of the shear-wave velocity profile, which is fundamental for evaluating the dynamic response of soil. Additionally, integrating these geophysical constraints with geotechnical and geological data will support the development of 3D subsurface models, improving both our understanding of the caldera’s structure and our capacity for seismic microzonation.

## Figures and Tables

**Figure 1 sensors-25-04787-f001:**
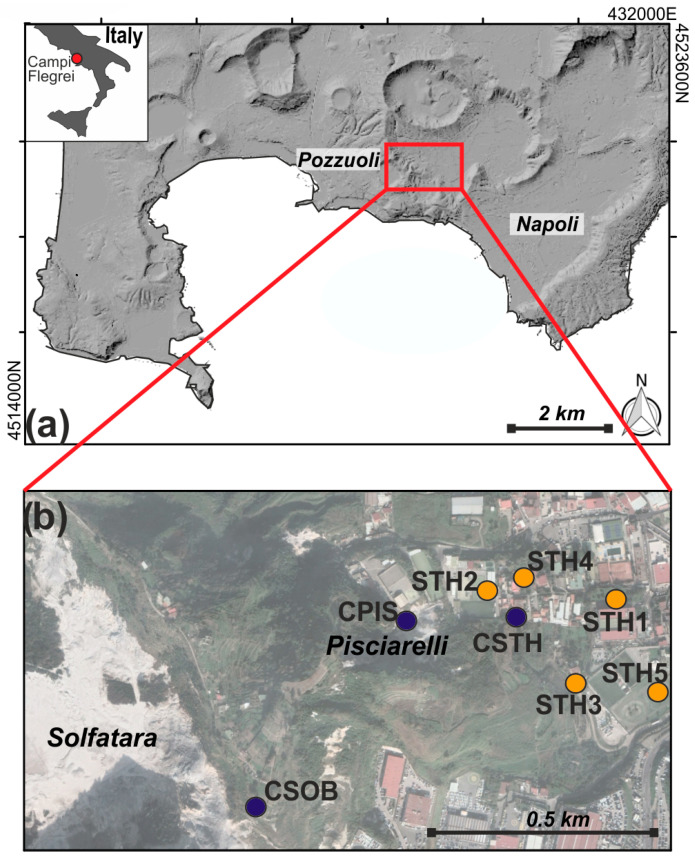
(**a**) Map of Campi Flegrei caldera (coordinates Datum WGS84, Geographic Latitude-N Longitude-E); (**b**) detail of the studied area, showing the location of the permanent stations of the INGV-OV monitoring network (blue circles) and the location of the temporary stations (orange circles).

**Figure 2 sensors-25-04787-f002:**
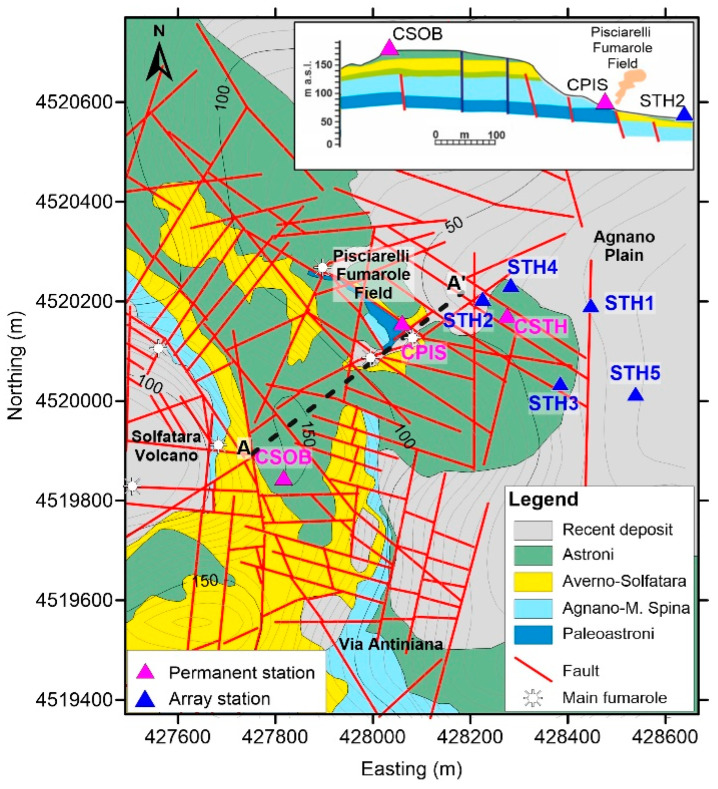
Simplified geological map of the area around CSTH station. The top right panel shows the simplified geologic section obtained along the A–A’ profile, on which the intercepted seismic stations are shown (modified after Isaia et al. [31]).

**Figure 3 sensors-25-04787-f003:**
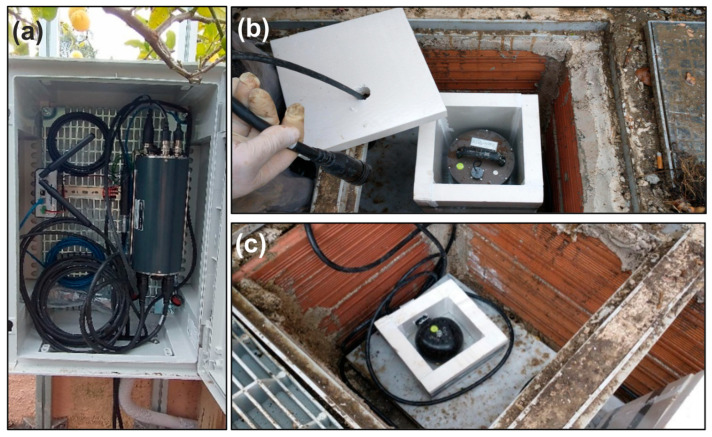
(**a**) Affinity Guralp digitizer, (**b**) Guralp ESPC velocimeter-120s, and (**c**) Guralp Fortis accelerometer.

**Figure 4 sensors-25-04787-f004:**
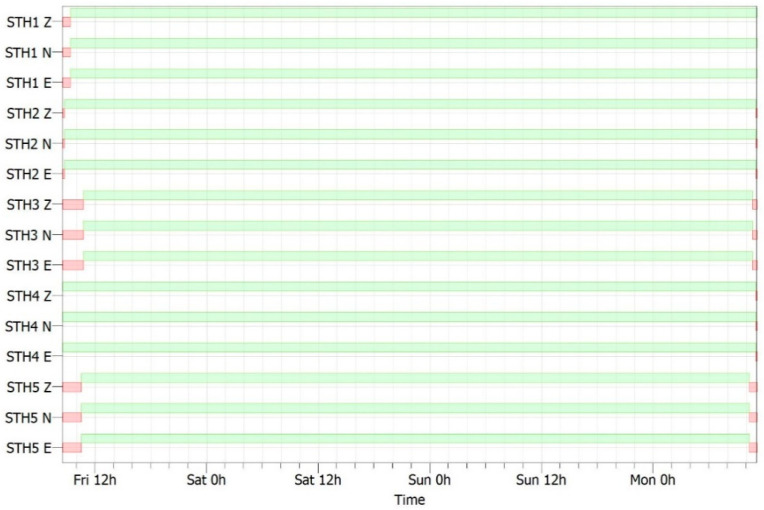
Chronogram of the performance of the array stations. Green bands show the recording period while red bands indicate no acquisition.

**Figure 5 sensors-25-04787-f005:**
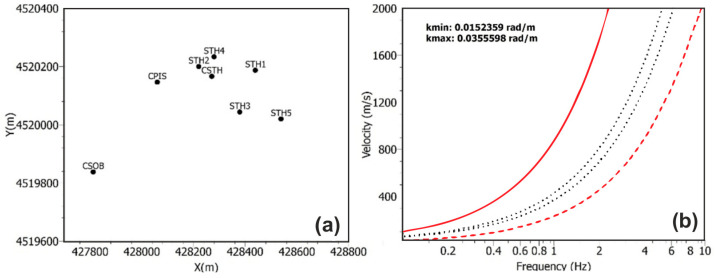
(**a**) Two-dimensional geometry of the array composed of the five short-period temporary stations (STH*) and the three broad-band stations of the permanent network at the Campi Flegrei (CSTH, CPIS, and CSOB). (**b**) Array resolution curves: kmin (dashed thick line), kmax (dotted line), kmax/2 = kmix/2 (dashed lines).

**Figure 6 sensors-25-04787-f006:**
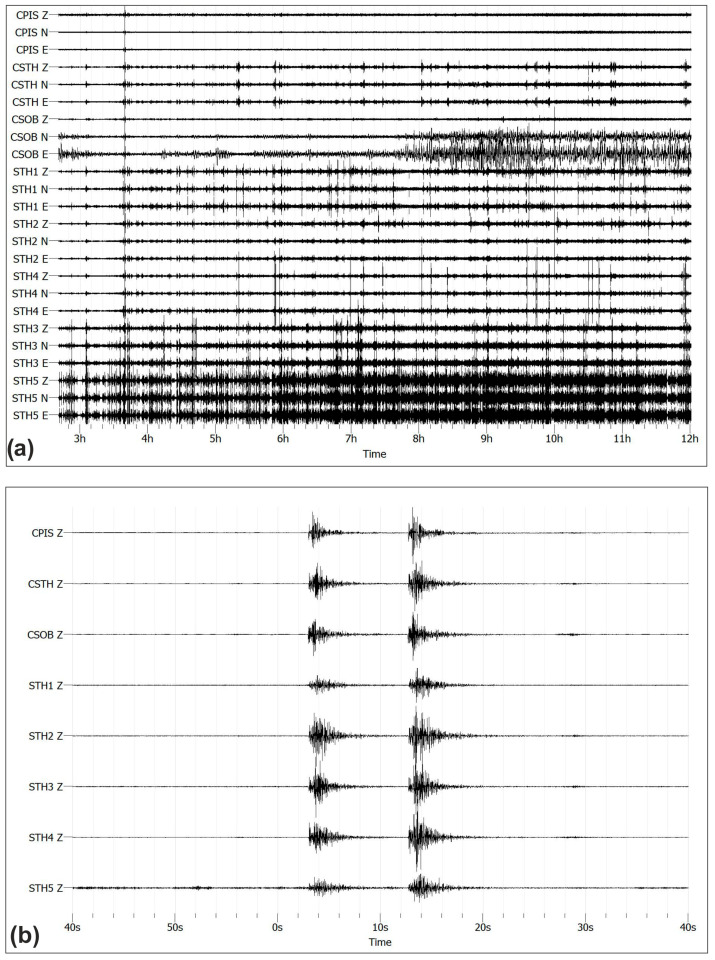
(**a**) Seismic traces in the vertical (V) and horizontal (E, N) components recorded on 05/28/2022 from 02:40:00 to 12:00:00 UTC at the five temporary stations (STH*) and at the three stations of the permanent Campi Flegrei network (CSTH, CSOB, and CPIS); (**b**) vertical traces of two local earthquakes of magnitude durations Md 0.2 and 0.7 recorded on 28 May 2022 at 22:23:02 and 22:23:12 UTC, respectively. Unfiltered signals of vertical components recorded at digital seismic stations are shown in the graph.

**Figure 7 sensors-25-04787-f007:**
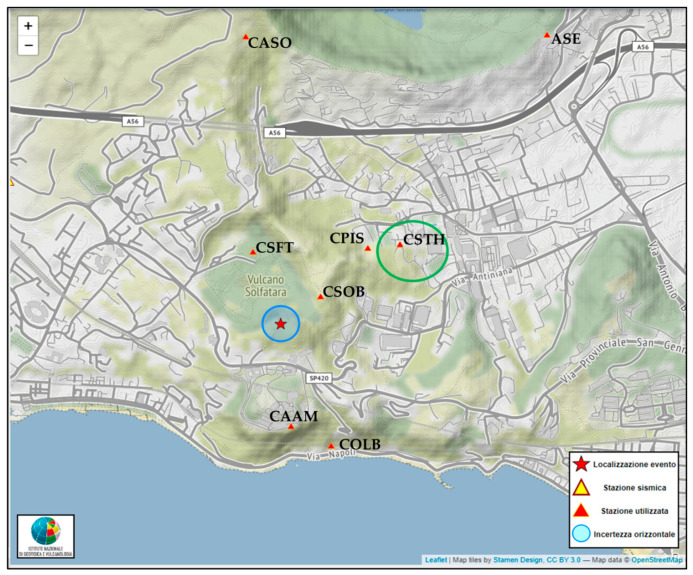
Location (red star) of the 0.2 and 0.7 *M_d_* events that occurred on 28 May 2022 at 22:23:12 UTC (the amplitude of the light blue circle identifies the location horizontal error), at a depth of 0.7 km a.s.l., latitude of 40.825298° N, and longitude of 14.141300° E (from https://terremoti.ov.ingv.it/gossip/flegrei/2022/event_15857.html). Circled in green is the study area for the CSTH station characterization. Red triangles represent the stations that were used for this location.

**Figure 8 sensors-25-04787-f008:**
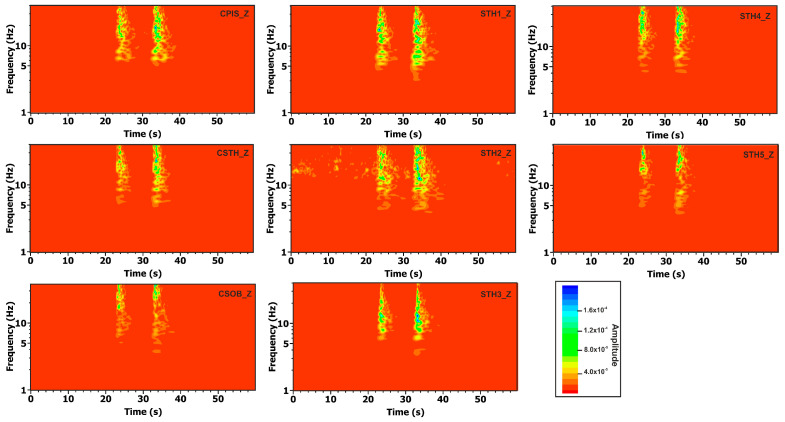
Spectrogram of the vertical components for the two seismic events with magnitude *M_d_* 0.2 and 0.7 that occurred at 22:23:02 and 22:23:12 UTC on 28 May 2022, respectively.

**Figure 9 sensors-25-04787-f009:**
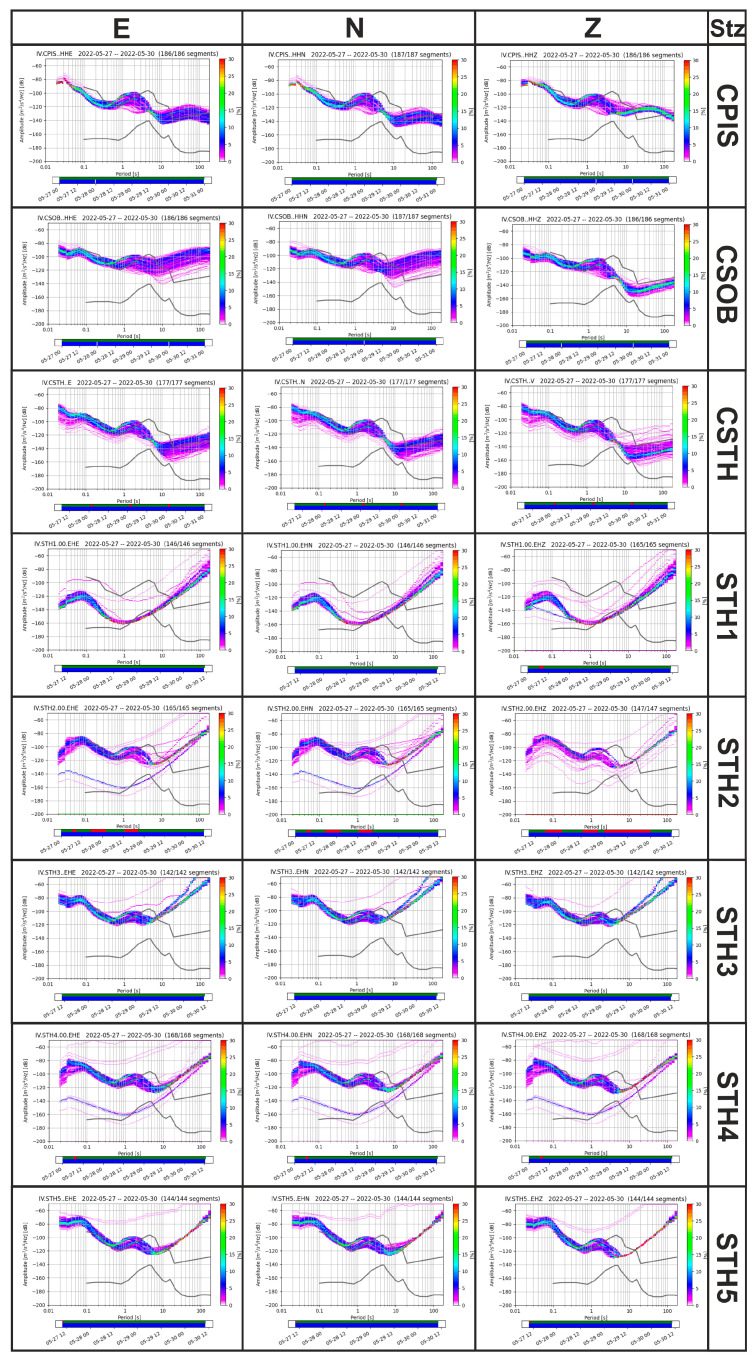
PSD calculated for all the stations in three directions E—N—Z, for all the datasets. The curves from [40] (grey continuous line) were used as comparison parameters.

**Figure 10 sensors-25-04787-f010:**
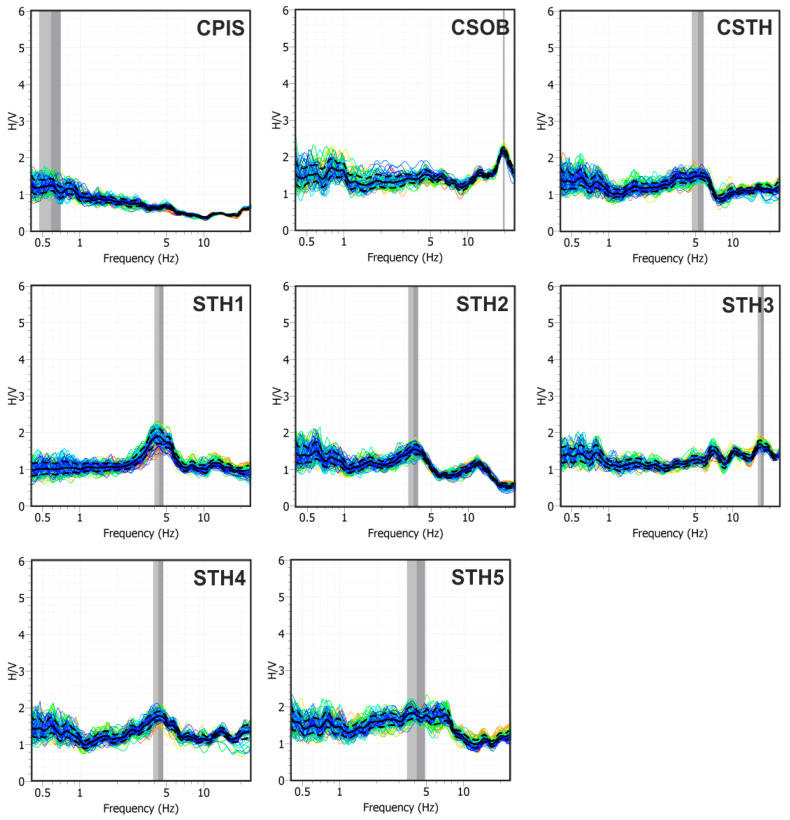
H/V spectral ratio curves. The solid black line represents the average of the calculated spectral ratios over the individual time windows, the dashed black lines represent the standard deviations, and the colored lines are the curves obtained over the individual analysis windows.

**Figure 11 sensors-25-04787-f011:**
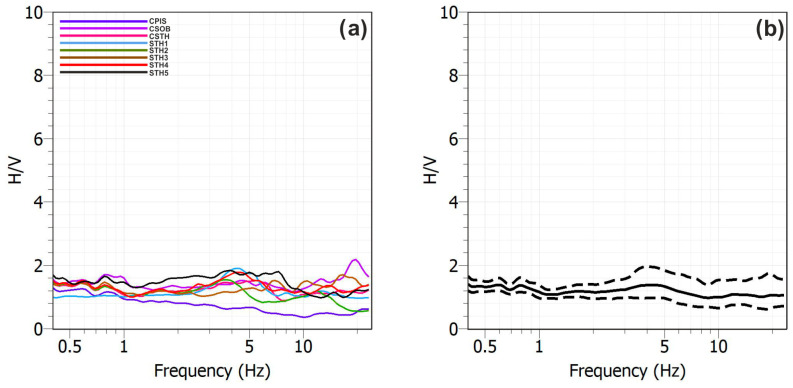
(**a**) Overlay of the average H/V curves obtained at the different sites and (**b**) their average: the solid black line represents the average of the spectral ratios calculated over the individual time windows, while the dashed lines represent the standard deviations.

**Figure 12 sensors-25-04787-f012:**
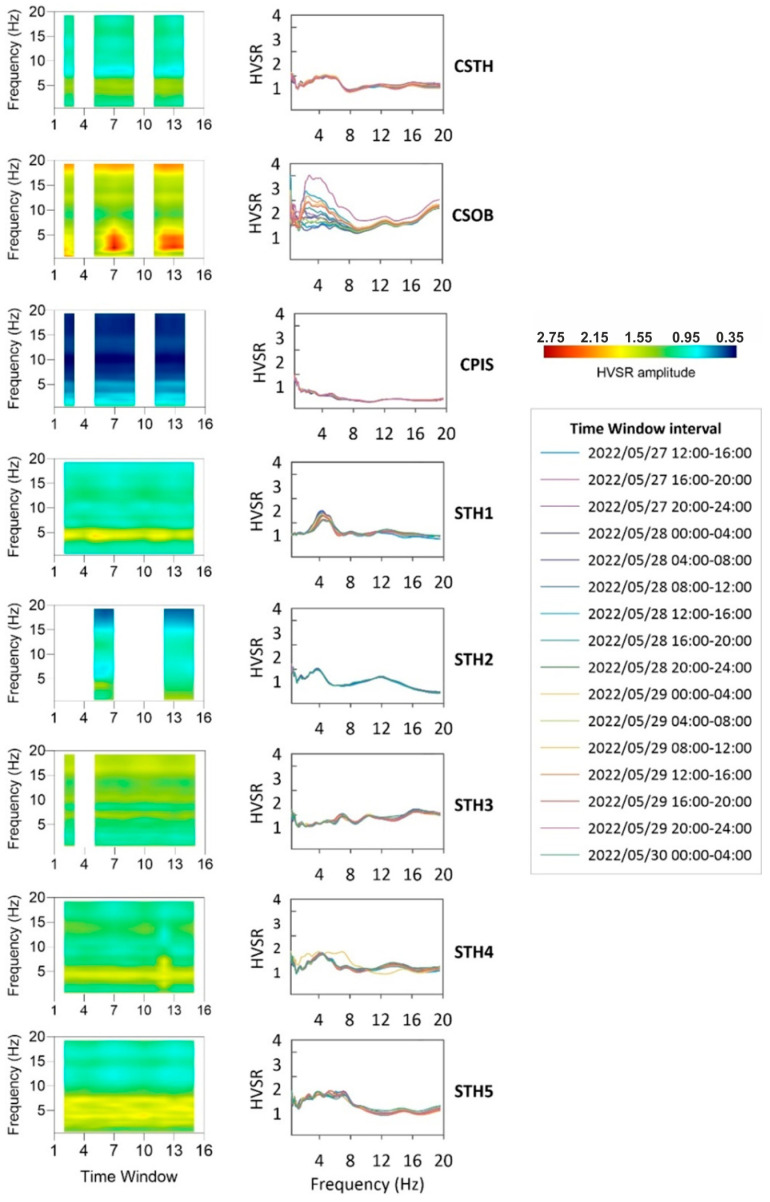
Temporal variations of the H/V ratios computed on the entire dataset; for each station considered, the change in time frequency is shown on the left (color scale at lower right) and the change in H/V ratio with frequency for the 16 4 h time intervals into which the dataset was divided on the right.

**Figure 13 sensors-25-04787-f013:**
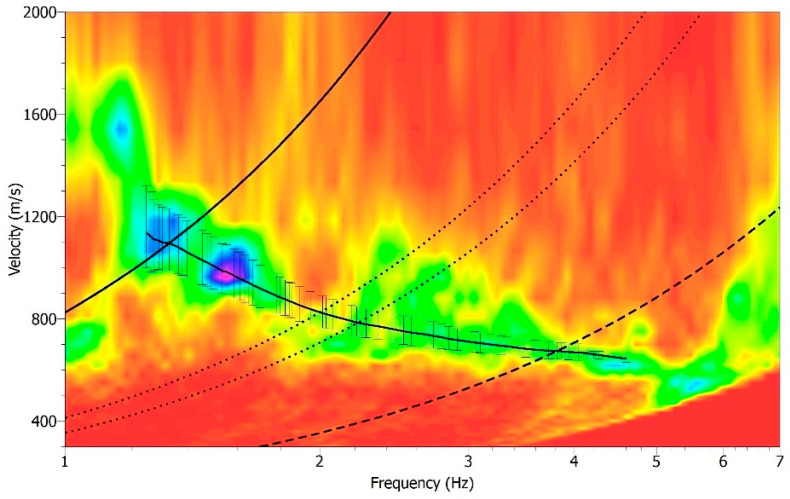
Dispersion curve-corresponding to the maximum spectral amplitude in the velocity-frequency domain (where purple is maximum amplitude and red the lowest), obtained through f-k analysis performed on the vertical component of the data. The black line represents the picked dispersion curve with corresponding error bars. Resolution limits are shown as follows: kmin (thick dashed line), kmax (thick solid line), and kmin/2 and kmax/2 (dotted lines).

**Figure 14 sensors-25-04787-f014:**
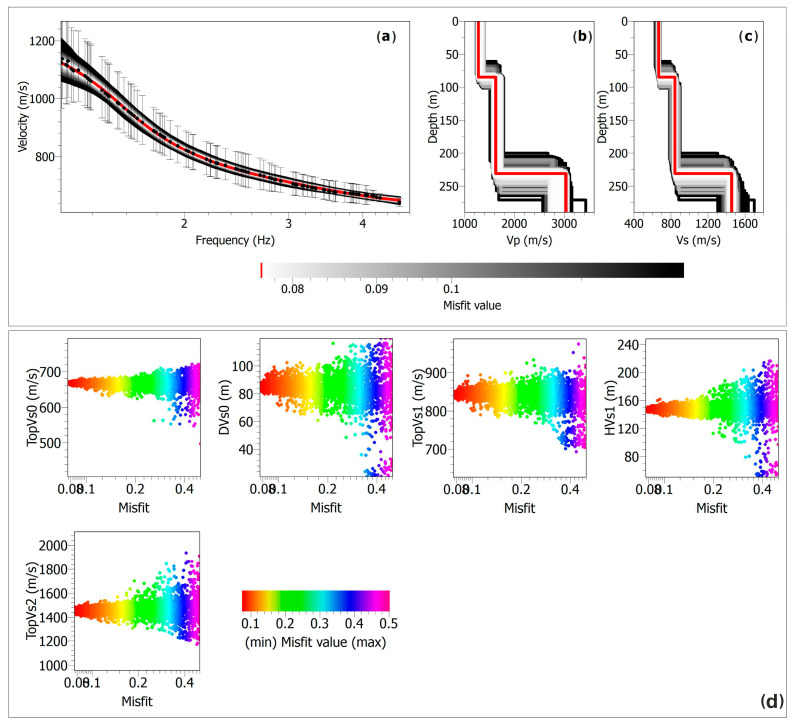
Results of the dispersion curve inversion. (**a**) Observed phase velocities (black dots) with error bars, minimum misfit model phase velocities (red line), and range of stable models (gray band). (**b**) Minimum misfit P-wave velocity model (red line) and range of stable models (gray band). (**c**) Minimum misfit S-wave velocity model (red line) and range of stable models (gray band). (**d**) Projection of all generated models with a minimum error below 0.5, shown as a function of the error. Vs0 and DVs0 denote the shear wave velocity and thickness of the first layer, Vs1 and HVs1 denote those of the second layer, and Vs2 denotes the shear wave velocity of the half-space. The misfit quantifies the discrepancy between the observed data and model predictions obtained from the inversion process.

**Table 1 sensors-25-04787-t001:** Geographic coordinates of the temporary and the permanent stations, with instrumental characteristics.

Site	Lat	Lon	Logistic	Acq.	Sensor	Cps	Start Registration	End Registration	Type of Soil
STH1	40.8296	14.1513	Installed inside the garden of the Altamira Complex	Lunitek Atlas	Lennartz 3DLite	100	05/27/2022 09:20:00	05/30/2022 11:10:00	Pyroclastic deposits
STH2	40.8297	14.1487	Installed next to the soccer fields immediately below Pisciarelli	Lunitek Atlas	Lennartz 3DLite	100	05/27/2022 08:40:00	05/30/2022 11:00:00	Pyroclastic deposits
STH3	40.4942	14.0903	Installed in dirt road outside Ariel Car Pozzuoli	Guralp Affinity	Lennartz 3DLite	100	05/27/2022 10:50:00	05/30/2022 10:15:00	Pyroclastic deposits
STH4	40.4948	14.0858	Installed in a private garden in the same complex as STH	Lunitek Atlas	Lennartz 3DLite	100	05/27/2022 08:30:00	05/30/2022 11:00:00	Pyroclastic deposits
STH5	40.4941	14.0909	Installed in a flowerbed at Ariel Car Pozzuoli	Guralp Affinity	Lennartz 3DLite	100	05/27/2022 10:30:00	05/30/2022 10:20:00	Pyroclastic deposits
CSTH	40.8294	14.1493	Solfatara Tennis Hotel	Guralp Affinity	Lennartz 3DLite	100	06/04/2022 00:00:00	Still working	Pyroclastic deposits
CPIS	40.8292	14.1468	Campi Flegrei—Fumarola Pisciarelli	GILDA	CMG-40T-60S	100	29/01/2010	May 2023	Pyroclastic deposits
CSOB	40.8264	14.1439	Solfatara Bordo Cratere	GILDA	CMG-40T-60S	100	11/07/2021	Still working	Pyroclastic deposits

**Table 2 sensors-25-04787-t002:** Parameters used in the antitrigger module for selection of windows for calculation of spectra (GEOPSY Software, 3.4.1 version, [44]).

Parameters	Values
STH length	1 s
LTA length	30 s
Minimum STA/LTA	0.20
Maximum STA/LTA	2.50
Window length	250 s

## Data Availability

Data used in this work are available at doi:10.5281/zenodo.15298023.

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
