# Peer review of "First Steps Towards Site Characterization Activities at the CSTH Broad-Band Station of the Campi Flegrei’s Seismic Monitoring Network (Italy)"

_sensors, 2025, doi:10.3390/s25154787_

Round 1

Reviewer 1 Report

Comments and Suggestions for Authors

Thank you for your contribution to Sensors. This manuscript presents a comprehensive seismic site characterization of the CSTH broad-band seismic station in the Campi Flegrei caldera. By integrating spectral analysis techniques—including H/V spectral ratio, PSD analysis, and f-k analysis—the authors offer a detailed assessment of local site effects. Overall, the manuscript is well-written and presents valuable results. However, in my opinion, the manuscript requires minor revisions to better explain the methodological choices, enhance scientific clarity, and improve visual presentation.

Below are my comments:

  1. The authors apply three spectral analysis techniques: Power Spectral Density (PSD), Horizontal-to-Vertical (H/V) spectral ratio, and f-k array processing to evaluate site response. Please elaborate on the rationale behind selecting these specific methods. What are the advantages of using these techniques, especially in the context of Campi Flegrei's complex geological setting?

  1. While the analysis is technically sound, the manuscript currently reads more like a report than a scientific paper. The authors should clearly articulate the novel insights or findings that result from combining the three techniques. What new understanding has been achieved through this integrative approach?

  1. The manuscript does not discuss whether environmental noise was filtered or mitigated prior to analysis. Were any noise-removal or pre-processing steps applied to improve signal quality? If so, please describe the methodology. If not, please justify.

  1. Lines 38–39, The manuscript mentions that there are limitations, but does not specify what they are. Please clearly state the limitations.

  1. Figure 5b, suggestion that using colored lines to better distinguish the curves in the plot.

  1. Figure 6a, the station STH5 shows a significantly stronger noise amplitude. Please clarify the possible reasons, could it be related to instrumentation, site effects, or local noise sources?

  1. Figure 8, the axis labels and station name are too blurry to read.

  1. Figure 10, it is difficult to distinguish between the solid and dashed black lines.

  1. line 403, “Only the CSOB station displays a peak centered around 15 Hz, with an amplitude slightly exceeding the value of 2”, Please explain the physical significance of this peak exceeding a value of 2. Does this imply local amplification due to impedance contrast or a specific subsurface feature?

  1. Reference [19] has a capitalization error. Please ensure all references follow the correct formatting style.

Author Response

The authors apply three spectral analysis techniques: Power Spectral Density (PSD), Horizontal-to-Vertical (H/V) spectral ratio, and f-k array processing to evaluate site response. Please elaborate on the rationale behind selecting these specific methods. What are the advantages of using these techniques, especially in the context of Campi Flegrei's complex geological setting?

We thank the reviewer for this valuable comment. In the revised version of the manuscript, we have clarified the scientific contribution of our integrative approach. Specifically, we emphasize how the combined use of PSD, H/V spectral ratio, and f-k analysis allows for a multi-scale and complementary characterization of the seismic site response in a geologically complex setting like the Campi Flegrei caldera. See lines 514-521.

While the analysis is technically sound, the manuscript currently reads more like a report than a scientific paper. The authors should clearly articulate the novel insights or findings that result from combining the three techniques. What new understanding has been achieved through this integrative approach?

 Thank you for the suggestion, the reviewer is total right. We tried to clarify the new understanding. See the revised conclusions.

The manuscript does not discuss whether environmental noise was filtered or mitigated prior to analysis. Were any noise-removal or pre-processing steps applied to improve signal quality? If so, please describe the methodology. If not, please justify.

We thank the reviewer for this important observation. In our analysis, no explicit filtering or removal of anthropogenic seismic noise was applied prior to processing. This choice is motivated by the nature of the study, which aims to characterize ambient seismic noise and its use for site response analysis, rather than to isolate specific transient seismic signals.

Lines 38–39, The manuscript mentions that there are limitations, but does not specify what they are. Please clearly state the limitations.

Thank you for the observation. We tried to clarify it, adding some lines 38-41: “[…] particularly in complex geological and topographical contexts, where slope-derived proxies may misrepresent site conditions—such as in volcanic terrains or sedimentary basins with low relief—and may suffer from regional biases due to calibration constraints and the inability to capture subsurface heterogeneities or basin structure”.

Figure 5b, suggestion that using colored lines to better distinguish the curves in the plot.

Thank you for the suggestion, we modified Figure 5b.

Figure 6a, the station STH5 shows a significantly stronger noise amplitude. Please clarify the possible reasons, could it be related to instrumentation, site effects, or local noise sources?

Thank you for your comment, we clarified it, lines 240-241: “[…] STH5 is located in a car park, so the greater amplitude of the noise can be attributed to local anthropogenic sources.”

Figure 8, the axis labels and station name are too blurry to read.

Thank you for your comment, we modified Figure 8.  

Figure 10, it is difficult to distinguish between the solid and dashed black lines.

Thank you for your comment, we modified Figure 10.  

line 403, “Only the CSOB station displays a peak centered around 15 Hz, with an amplitude slightly exceeding the value of 2”, Please explain the physical significance of this peak exceeding a value of 2. Does this imply local amplification due to impedance contrast or a specific subsurface feature?

Thank you for the comment, we explained it better in the text, lines 415-420: “[…] CSOB is probably localised above a local impedance contrast, although not particularly marked, related to the presence of buried rigid or hardened levels of loose or poorly consolidated volcanic material (Martino et al., 2025). Such configurations can produce modest amplification effects at high frequencies.

However, the low amplitude and high frequency of the peak suggest little marked and/or localised surface contrast, indicating a small local stratigraphic inhomogeneity.”

Reference [19] has a capitalization error. Please ensure all references follow the correct formatting style.

Thank you for the comment, we modified it.

Reviewer 2 Report

Comments and Suggestions for Authors

Local site conditions significantly influence seismic data analysis, particularly in seismic monitoring tasks. In the work, standard seismic noise level estimates are supplemented by two established techniques based on microseism analysis. The first, the H/V ratio method, requires only single-station recordings. The second method analyzes data from a network of short-period stations deployed near a broadband station. The resulting spectral characteristics — the H/V ratio and Rayleigh wave velocity — provide critical insights into the structure of the upper rock layers beneath the station, down to depths of approximately 10 km. Overall, the research quality and its presentation are commendable. However, one minor suggestion: the results could be complemented with the depth-dependent transverse wave velocity derived from the inversion of the dispersion curve.

Author Response

Local site conditions significantly influence seismic data analysis, particularly in seismic monitoring tasks. In the work, standard seismic noise level estimates are supplemented by two established techniques based on microseism analysis. The first, the H/V ratio method, requires only single-station recordings. The second method analyzes data from a network of short-period stations deployed near a broadband station. The resulting spectral characteristics — the H/V ratio and Rayleigh wave velocity — provide critical insights into the structure of the upper rock layers beneath the station, down to depths of approximately 10 km. Overall, the research quality and its presentation are commendable. However, one minor suggestion: the results could be complemented with the depth-dependent transverse wave velocity derived from the inversion of the dispersion curve.

We thank you a lot the reviewer for the positive comments and for the suggestion. We inverted the dispersion curve and inserted the model as Figure 14.

Reviewer 3 Report

Comments and Suggestions for Authors

Presented in this paper is the result of a preliminary effort conducted by the authors in an area rich in volcanic-seismic activity. The paper offers information and can be published to SENSORS after some revision with respect to: 1) references and 2) the description of the procedure followed when methods of refs. 43-46 are employed. In particular, the authors should mention only the most important references and not 100 ones. This is not a review paper. I suggest to reduce references to about 50 or less. Please keep in mind that those references that are really needed should be mentioned. Moreover, the authors should be more specific with respect to the implementation of the methods presented in refs. 43-46. Thus, section 4 should be carefully revised.

Author Response

Presented in this paper is the result of a preliminary effort conducted by the authors in an area rich in volcanic-seismic activity. The paper offers information and can be published to SENSORS after some revision with respect to: 1) references and 2) the description of the procedure followed when methods of refs. 43-46 are employed. In particular, the authors should mention only the most important references and not 100 ones. This is not a review paper. I suggest to reduce references to about 50 or less. Please keep in mind that those references that are really needed should be mentioned. Moreover, the authors should be more specific with respect to the implementation of the methods presented in refs. 43-46. Thus, section 4 should be carefully revised.

We thank you the reviewers for the suggestions. We reduced a lot the references, however we think that the ones left are fundamental for the paper. We have revised Chapter 4 as requested, and we hope the new version is clearer. We remain available for any further suggestions that could help improve the work.

Round 2

Reviewer 3 Report

Comments and Suggestions for Authors

The authors are encouraged to prepare a revise version in which they ought to:

  • Reduce the number of references to about 50-55. I do not think that citing to many references is a benefit for the paper that essentially constitutes a case-study.
  • Mention soil conditions (perhaps by using the velocity of shear waves to a depth of 30m) in Table 1.
  • Explain the use of coda duration magnitude instead of surface wave magnitude (Rayleigh waves are employed in f-k technique).
  • Be more precise regarding noise filtering procedures (sections 4-2 & 4.3). Which tests were conducted and dictated removal of frequencies below 0.4 Hz and beyond 20 Hz? Also, why b is set equal to 40 (line 407)?
  • Explain the selection/use of geometric mean in section 4.3
  • Mention if severe seismic soil amplification is anticipated in the area studied and what implications may introduce to the seismic hazard of the area (in terms of peak-ground acceleration or peak-ground velocity).

Author Response

  • Reduce the number of references to about 50-55. I do not think that citing to many references is a benefit for the paper that essentially constitutes a case-study.

The reviewer is completely right, we have reduced the number of  references.

  • Mention soil conditions (perhaps by using the velocity of shear waves to a depth of 30m) in Table 1.

We thank the reviewer for the suggestion. We added the geological soil type that characterises each site in Table 1 to obtain an initial characterisation of the conditions at each site.At present, no site-specific studies providing VS30 values are available for the investigated area. As a result, current soil conditions in terms of seismic response parameters such as VS30 remain undetermined. However, future work is specifically aimed at addressing this gap. As stated in the manuscript, expanded data acquisition and the application of joint inversion techniques are planned to improve the resolution of the shallow shear-wave velocity profile. This will allow for a reliable estimation of VS30 and, consequently, a more accurate assessment of local soil amplification effects and dynamic response. The integration of these results with geological and geotechnical data will support the construction of detailed 3D subsurface models essential for seismic microzonation and hazard evaluation.

  • Explain the use of coda duration magnitude instead of surface wave magnitude (Rayleigh waves are employed in f-k technique).

The reviewer's point is well-founded. In Campi Flegrei caldera it is used the coda duration magnitude (https://doi.org/10.1038/s41598-021-86506-6 or https://doi.org/10.1785/0120070131) in order to avoid potential errors due to inaccurate event depth estimates, due to the very shallow nature of its seismicity. In fact, earthquakes in this area typically occur at depths ranging from the surface to about 4 km, with over 95% of events having MD < 1.0, as reported in the INGV Monthly Bulletins (https://www.ov.ingv.it/index.php/monitoraggio-e-infrastrutture/bollettini-tutti/bollett-mensili-cf/anno-2025-3) . Given these conditions, a reduced exclusion threshold—on the order of 2–3 km—could be more appropriate. Therefore, magnitude estimation methods for volcanic areas such as Campi Flegrei must be adapted to the unique seismological and network configuration of the site, possibly favoring alternative approaches less sensitive to depth uncertainties, such as coda duration magnitude.

  • Be more precise regarding noise filtering procedures (sections 4-2 & 4.3). Which tests were conducted and dictated removal of frequencies below 0.4 Hz and beyond 20 Hz? Also, why b is set equal to 40 (line 407)?

We thank the reviewer for the request. In our study, we decided to follow Nakamura’s method precisely, that’s why we used a bandwidth coefficient (b) set to 40 and applied a  band-pass filter bwteween 0.4–20 Hz, which is a reasonable standard for monitoring in urban volcanic areas, especially for small magnitude events and continuous signals. Frequencies below 0.4 Hz are typically dominated by long-period ambient noise (such as wind and barometric variations), and frequencies above 20 Hz, where the signal-to-noise ratio for volcanic signals,  including VT, LP and tremor events, tends to decrease dramatically, making the analysis less reliable. I sensori utilizzati per le stazioni STH* sono a corto periodo (freq. Propria 1Hz) che ci permettono di investigare fino a freq di 0.4-0.5 Hz il segnale registrato. Le stazioni della rete sismica permanente sono stazioni broad-band, che permettono di vedere al di sotto di 0.4 Hz, fino a 0.01 Hz; di conseguenza, per avere un dataset uniforme abbiamo scelto di applicare il filtro a 0.4 per non avere segnali distorti dalla risposta strumentale per le stazioni dell’array. No other filtering procedures were applied.

  • Explain the selection/use of geometric mean in section 4.3

We thank the reviewer for raising a valid request. The geometric mean is preferred over the arithmetic or quadratic mean in H/V analysis, as it avoids dominance of one horizontal component over the other, which may occur when seismic energy is not uniformly distributed across the horizontal directions. Additionally, the geometric mean is invariant with respect to the orientation of the horizontal axes, ensuring robustness and comparability across different stations. This approach is consistent with standard practices in ambient noise analysis (e.g., SESAME, 2004). According to Nakamura’s technique, the H/V method reflects average soil properties, which are assumed to represent the variation of soil behavior in both horizontal directions. Therefore, using the geometric mean of the two horizontal components provides a more representative and physically meaningful input for the H/V spectral ratio."Bard, P. Y., & Participants, S. E. S. A. M. E. (2004, August).

  • Mention if severe seismic soil amplification is anticipated in the area studied and what implications may introduce to the seismic hazard of the area (in terms of peak-ground acceleration or peak-ground velocity).

We thank the reviewer for raising a valid request. The results obtained show no marked resonance peaks in the H/V ratios, suggesting that the area studied is not subject to significant seismic ground amplification at the frequencies analysed. However, “although our study does not reveal significant amplification effects related to subsurface stratigraphy, very high peak ground acceleration (PGA) values have been recorded in the Campi Flegrei area. The observed values are extremely variable and highly localized near the epicentral region. Several studies have shown that earthquakes of similar magnitude can generate markedly different acceleration levels. This variability can be partially attributed to the location and depth of the seismic event (https://www.ingegneriasismicaitaliana.com/articoli-tecnici/ingv-isi-campi-flegrei-nuove-prospettive-nella-valutazione-del-rischio-sismico-attraverso-approcci-integrati-di-monitoraggio-sismico-e-strutturale, Italian site)”, lines 557-564.